# Lung Adenocarcinoma Cell Sensitivity to Chemotherapies: A Spotlight on Lipid Droplets and *SREBF1* Gene

**DOI:** 10.3390/cancers14184454

**Published:** 2022-09-14

**Authors:** Anna Ricarda Gründing, Marc A. Schneider, Sarah Richtmann, Mark Kriegsmann, Hauke Winter, Beatriz Martinez-Delgado, Sarai Varona, Bin Liu, David S. DeLuca, Julia Held, Sabine Wrenger, Thomas Muley, Michael Meister, Tobias Welte, Sabina Janciauskiene

**Affiliations:** 1Department of Respiratory Medicine, Member of the German Center for Lung Research (DZL), Biomedical Research in Endstage and Obstructive Lung Disease Hannover (BREATH), Hannover Medical School, 30625 Hannover, Germany; 2Translational Research Unit, Thoraxklinik at Heidelberg University Hospital, 69126 Heidelberg, Germany; 3Translational Research Center Heidelberg (TLRC), Member of the German Center for Lung Research (DZL), 69120 Heidelberg, Germany; 4Division of Systems Biology of Signal Transduction, German Cancer Research Center (DKFZ), 69120 Heidelberg, Germany; 5Institute of Pathology, Heidelberg University Hospital, 69120 Heidelberg, Germany; 6Department of Surgery, Thoraxklinik at Heidelberg University Hospital, 69126 Heidelberg, Germany; 7Department of Molecular Genetics, Institute of Health Carlos III, Center for Biomedical Research in the Network of Rare Diseases (CIBERER), 28220 Majadahonda, Spain; 8Bioinformatics Unit, Institute of Health Carlos III, 28220 Majadahonda, Spain

**Keywords:** lung adenocarcinoma, chemotherapy, lipid droplets, SREBF1, neutrophils

## Abstract

**Simple Summary:**

The accumulation of lipid droplets (LDs) and the high expression of genes involved in LD formation, such as *SREBF1* (sterol regulatory element binding transcription factor 1), are attributed to cancer cell resistance against anticancer drugs and poor prognosis. We assessed lung cancer cells with and without LDs for their sensitivity to chemotherapeutics cisplatin and etoposide. In either serum-free basal medium or inflammatory supernatants generated during neutrophil degranulation in vitro, both drugs strongly reduced *SREBF1* expression, which did not parallel with LD formation and cell sensitivity to chemotherapeutics. Nevertheless, under basal conditions, SREBF1 expression in cancer cells correlated with LD levels, and the lower expression of *SREBF1* in tumors than in adjacent nontumor tissues showed a prognostic value for overall better survival of patients with non-small-cell lung cancer. Strategies targeting lipid metabolism in cancer are promising therapeutic and/or diagnostic approaches.

**Abstract:**

To explore the relationship between cancer cell SREBF1 expression, lipid droplets (LDs) formation, and the sensitivity to chemotherapies, we cultured lung adenocarcinoma cells H1299 (with LD) and H1563 (without LD) in a serum-free basal medium (BM) or neutrophil degranulation products containing medium (NDM), and tested cell responses to cisplatin and etoposide. By using the DESeq2 Bioconductor package, we detected 674 differentially expressed genes (DEGs) associated with NDM/BM differences between two cell lines, many of these genes were associated with the regulation of sterol and cholesterol biosynthesis processes. Specifically, *SREBF1* markedly declined in both cell lines cultured in NDM or when treated with chemotherapeutics. Despite the latter, H1563 exhibited LD formation and resistance to etoposide, but not to cisplatin. Although H1299 cells preserved LDs, these cells were similarly sensitive to both drugs. In a cohort of 292 patients with non-small-cell lung cancer, a lower *SREBF1* expression in tumors than in adjacent nontumor tissue correlated with overall better survival, specifically in patients with adenocarcinoma at stage I. Our findings imply that a direct correlation between *SREBF1* and LD accumulation can be lost due to the changes in cancer cell environment and/or chemotherapy. The role of LDs in lung cancer development and response to therapies remains to be examined in more detail.

## 1. Introduction

The multifaceted heterogeneity of cancers is one of the most challenging factors limiting the success of cancer therapies. Different levels of cancer heterogeneity have been recognized, including the rates of genetic alterations and changes in metabolic activity, damage/repair and apoptotic signaling, and lipid metabolism pathways, which are influenced by cellular and noncellular components of the tumor microenvironment (TME) as well as by the individual genetic background [1,2].

The composition of the TME is a factor strongly affecting cancer cells, including the formation of lipid droplets (LDs), lipid metabolism, and drug resistance [3]. Cancer cells utilize LDs to ensure energy storage but also to modulate autophagy and minimize stress responses. Accumulating evidence supports a relationship between tumor development and LD mobilization [4]. Sterol regulatory element-binding factors (SREBFs) are key regulators of lipid homeostasis [5], and specifically, the upregulation of *SREBF1* gene seems to trigger LD accumulation and tumor growth [3,6]. Hence, targeting *SREBF1* overexpression and LD formation represents one of the promising strategies in cancer therapy. Up to date, different drugs, such as celecoxib, known as COX-2 inhibitor, or C75, an inhibitor of fatty acid synthase [7,8,9], have been shown to affect LD formation.

There are many studies showing that tumor-infiltrating neutrophils secrete a wide range of cytokines/chemokines, proteolytic enzymes, and reactive oxygen species [10], and there is a positive correlation between tumor-associated neutrophils and poor patient outcome [11,12]. In contrast, other studies report that neutrophils can acquire an antitumor phenotype [13,14]. There are also data suggesting that initially, neutrophils may express antitumor effects whereas at later stages of cancer, they can undergo a transition towards a protumorigenic phenotype [15]. Hence, neutrophils seem to exhibit pleiotropic roles during tumor initiation, development, and progression. So far, neutrophil’s role in the regulation of LD formation and *SREBF1* expression has been studied much less.

In this study, we aimed to investigate the responses of lung cancer cells to chemotherapeutics, specifically cisplatin and etoposide, under cell culture in the serum-free basal medium or in a medium containing neutrophil degranulation substances. We specifically focused on *SREBF1* gene expression and LD status in cancer cells in vitro, and on the *SREBF1* gene expression in a tumor and adjacent healthy tissues of lung adenocarcinoma and squamous cell carcinoma patients.

## 2. Materials and Methods

### 2.1. Chemotherapeutic Drugs

The platinum-based chemotherapeutic cisplatin (Merck, Darmstadt, Deutschland) induces a cross-linking of DNA strands (intra- and interstrand cross-linking), blocks DNA synthesis, and leads to apoptosis [16]. Etoposide (Sigma-Aldrich, Taufkirchen, Germany) blocks topoisomerase II and DNA ligation, inducing cell cycle arrest and apoptosis induction [17].

### 2.2. Neutrophil Isolation and Degranulation

Human neutrophils were isolated from healthy volunteer blood by using Polymorphprep (Axis-Shield PoC AS, Oslo, Norway) according to the manufacturer’s recommendations as described elsewhere [18]. Isolated neutrophils were counted with Cellometer Auto T4 (Nexcelom, Lawrence, MA, USA) and suspended in RPMI 1640 at a final concentration of 6 × 10 ^6^ cells/mL. Neutrophils were degranulated for 10 min at 30 °C using an ultrasound bath at the highest speed (EMMI D60, EMAG, Mörfelden-Walldorf, Germany). After degranulation, cells were pelleted at 500× *g* for 10 min at RT. Cell-free supernatants were collected, and degranulation was verified by the quantification of human myeloperoxidase (MPO) levels using DuoSet ELISA (detection rate: 62.5–4000 pg/mL; R&D Systems, Minneapolis, Minnesota, USA) and by measuring protease activity using Novex 10% zymogram plus (Gelatin) gels (Thermo Fisher Scientific) (Appendix A). Freshly prepared neutrophil degranulation medium (NDM) was used for cancer cell cultures.

### 2.3. Cell Culture and In Vitro Experiments

NSCLC NCI-H1299 (ATCC CRL-5803), NCI-H1563 (ATCC CRL-5875), NCI-H1437 (ATCC CRL-5872), NCI-H661 (ATCC HTB-183), NCI-H1573 (ATCC CRL-5877), and NCI-H1975 (ATCC CRL-5908) cell lines were purchased from ATCC (Virginia, MA, USA). Cells were cultured in RPMI 1640 with 5 or 10% fetal bovine serum (Gibco Thermo Fisher Scientific, Waltham, MA, USA) at 37 °C and 5% CO_2_. During the experiments, cells were cultured in serum-free RPMI 1640 (basal medium, BM) or neutrophil degranulation medium prepared with RPMI 1640 (NDM). Cells were treated with 100 µM etoposide or 20 µg/mL cisplatin for 48 h.

### 2.4. RNA Isolation

RNA isolation was performed with an RNeasy Mini Kit (Qiagen, Venlo, The Netherlands) according to the manufacturer’s recommendations. RNA concentration was measured with Nanodrop (Thermo Fisher Scientific, Waltham, MA, USA) and the quality of total RNA was assessed by 1% agarose gels or by utilizing Agilent 2100 bioanalyzer and Agilent RNA 6000 Nano Kit (Agilent Technologies, Boeblingen, Germany).

### 2.5. Transcriptome Analysis (RNA-Seq)

Libraries were prepared from 200 ng RNA of each cell line with TruSeq Stranded mRNA Kit (Illumina) according to the manufacturer protocol as described previously [19].

### 2.6. RNA-Seq Data Analysis

RNA sequencing data in the FASTQ.gz form were mapped to the human genome GRCh38 and were aligned to the human genome (UCSC mm10) by STAR v2.4.0.1 [20] aligner to get the corresponding BAM files. Uniquely mapped reads were counted by HT-seq v0.11.1 [21] to quantify the expression levels in raw counts. The normalization and differential expression analysis were performed on raw counts using the R package DESeq2 v1.32.0 based on default settings [22]. Interaction terms were added to the design matrix to increase the model’s sensitivity in detecting the differences in treatment effects between cell lines. The normalization and differential expression analysis were performed on raw counts from RNA-seq using the R package DESeq2 v1.32.0 (*p*-values attained by the Wald test were corrected for multiple testing using the Benjamini and Hochberg method using the default and recommended settings from DESeq2) [22]. In a first analysis, we assessed the differential expression caused by the choice of medium in each cell line independently. To assess whether some differences could be shown to be cell-line-dependent, an interaction term was added to the model. This ensured that the reported differences in medium effect across the two cell lines (i.e., “cell-line-dependent medium effects”) were rigorously statistically significant.

Multiple testing corrections were processed with the default method in DESeq2. Differentially expressed genes (DEGs) were defined as those with an adjusted *p*-value smaller than 0.05. A gene set enrichment analysis (GSEA) was done on DEGs using the R package Enrichr Version [3.0] (created by W. Jawaid from New York, NY, USA) [23]. Significant gene ontology biological process (GO BP) terms and KEGG pathways were defined as gene set results with an adjusted *p*-value smaller than 0.05. The normalized gene expression levels and DEG results were visualized using R and the related packages, including ggplot2 Version [3.3.5] (created by H. Wickham et al. from Palo Alto, CA, USA) [24], ggrepel Version [0.9.1] (created by K. Slowikowski from Boston, MA, USA) [25], and pheatmap Version [1.0.12] (created by R. Kolde from Tartu, Estonia) [26].

### 2.7. CDNA Synthesis and Quantitative Real-Time PCR Analysis

Total RNA from cell lines was isolated using RNeasy Mini Kit (Qiagen) following the manufacturer’s instructions. RNA was reverse-transcribed with High Capacity cDNA Reverse Transcription Kit and cDNA was amplified with the following Taqman gene expression assays (Applied Biosystems, Thermo Fisher Scientific, Waltham, MA, USA): *LDLR* (Hs01092524_m1), *HMGCS1* (Hs00266810_m1), *HMGCR* (Hs00168352_m1), *CDKN1A* (Hs00355782_m1), *SREBF1* (Hs01088691_m1), *SERPINE1* (Hs01126606_m1), *AGPAT2* (Hs00944961_m1), *THBS1* (Hs00962908_m1), *TOPBP1* (Hs00199775_m1), *WNT3* (Hs00902257_m1), and *POLR2A* (Hs00172187_m1). TaqMan Gene Expression Master Mix (Applied Biosystems) and fluorescence reader StepOnePlus Real-Time PCR Systems (Applied Biosystems) were employed according to the manufacturer’s protocol. *POLR2A* was used as a housekeeping gene, and the gene expression of a target gene relative to *POLR2A* was calculated as relative mRNA levels (ΔCt). Measurements were carried out in duplicate.

### 2.8. PE Annexin V Apoptosis Detection

Apoptosis was assessed with PE Annexin V Apoptosis Detection Kit (BD Pharmingen, San Diego, CA, USA) according to the manufacturer’s instructions. Cells negative for annexin V and 7-AAD (bottom left quadrant) were classified as “living” (Appendix A).

### 2.9. Metabolic Assay

The metabolic activity was determined with colorimetric-based CyQuant XTT Cell Viability Assay kit (Thermo Fisher Scientific, Bend, OR, USA) according to the supplier’s instructions. The specific absorbance was determined by the following formula: Specific Absorbance = (Abs 450 nm (Test) − Abs 450 nm (Blank)) − Abs 660 nm (Test).

### 2.10. Oil Red Staining

Cancer cells were grown on coverslips and fixed with 3% paraformaldehyde (Roth, Karlsruhe, Germany) prior to staining. Freshly prepared Oil red O working solution (Sigma-Aldrich) was used for LD and hematoxylin (Roth) for nucleus staining. Coverslips were mounted with Roti-Mount Aqua (Roth) before images were taken with a Leica DM750 (Leica, Wetzlar, Germany) equipped with a Leica ICC50 HD camera (Leica). The analysis of the area covered by LDs was performed by using ImageJ Fiji (https://imagej.net/Fiji/Downloads (accessed on 25 August 2022)). The area of the LDs is given as a percentage relative to the covered area of the cancer cells.

### 2.11. Dot Blots 

Cancer cells were scraped in ice-cold PBS containing a 0.1% protease inhibitor cocktail (Sigma-Aldrich, St. Louis, MO, USA) and sonicated for 6 s before centrifugation at 4 °C and 11,000× *g* for 15 min. A constant amount (22 µg) of the lysates was loaded onto 0.2 µm nitrocellulose membranes (GE healthcare life sciences, Freiburg, Germany). After drying for 2 h at room temperature, membranes were blocked with TBS containing 0.1% Tween 20 (TBST) and 5% bovine serum albumin for 1 h at room temperature and incubated overnight with rabbit polyclonal antihuman SREBF1 antibody (1:400, Abcam, Amsterdam, The Netherlands). The immune complexes were visualized using horseradish peroxidase conjugated secondary antibody (antirabbit 1:10,000, DAKO, Glostrup, Denmark) and developed with ECL Western blotting substrate (Bio-Rad, München, Germany). Images were taken with the Chemidoc touch imaging system (Bio-Rad, Hercules, CA, USA).

### 2.12. Lung Cancer Patient Tissue Samples

Patient tissue samples were obtained from Lung Biobank Heidelberg, a member of the accredited Tissue Bank of the National Center for Tumor Diseases (NCT) Heidelberg, the Biomaterial Bank Heidelberg, and the Biobank platform of the German Center for Lung Research (DZL). Tumor and matched distant (>5 cm) tumor-free lung tissue samples from 292 NSCLC patients who underwent therapy-naive resection for primary lung cancer at Thoraxklinik at University Hospital Heidelberg, Germany were collected between 2006 and 2011. The local ethics committees of the Medical Faculty Heidelberg and Hannover Medical School (S-270/2001 and 9155_BO_K_2020) approved the use of biomaterial and data. All patients included in the study signed an informed consent and the study performed according to the principles set out in the WMA Declaration of Helsinki. The same cohort of lung cancer patients was already published in a previous study [19]. Therefore, more information is provided elsewhere. Patient data are shown in Appendix A.

### 2.13. SREBF1 Expression Analysis in Tissues

Gene expression analysis was performed as described by Ercetin et al. [19]. The following primers and UPL were used for the detection of *SREBF1*:

*SREBF1* forward (UPL #77, 5′-CGCTCCTCCATCAATGACA-3”), *SREBF1* reverse (UPL #77, 5′-TGCGCAAGACAGCAGATTTA-3”), *ESD* forward (UPL#50, 5′-TCAGTCTGCTTCAGAACATGG-3′), *ESD* reverse (UPL#50 5′-CCTTTAATATTGCAGCCACGA-3′), *RPS18* forward (UPL#46, 5′-CTTCCACAGGAGGCCTACAC-3′), *RPS18* reverse (UPL#46, 5′-CGCAAAATATGCTGGAACTTT-3′).

The two genes *ESD* and *RPS18* were the most stable ones and therefore used for all patient analyses. The whole procedure is described in the manuscript from 2015 in Clinical Cancer Research. After testing a variety of housekeeper genes for their stability in lung cancer and normal patient tissues [27], we used Esterase D (*ESD*) and Ribosomal Protein S18 (*RPS18*) as housekeepers for the comparison of gene expression in tumor and nonmalignant samples, and the relative expression of the genes was calculated (ΔCt values). For the waterfall plots analyzing the fold change expression changes between tumor and normal tissues, 2^-ΔΔCt^ values were calculated as described previously [27].

### 2.14. Statistical Analysis

The patient data of qPCR analyses were statistically analyzed under the REMARK criteria [28] with SPSS 26.0 for Windows (IBM, Ehningen, Germany). The endpoint of the study was overall survival. Overall survival time was calculated from the date of diagnosis until the last date of contact or death. The cutoffs used for survival analyses were selected using the software tool “Cutoff-Finder” [29]. Multivariate survival analyses were performed using the Cox proportional hazards model. A univariate analysis of survival data was performed according to Kaplan and Meier. The nonparametric, unpaired Mann–Whitney test was used to test the significance between the patient groups. The nonparametric Wilcoxon matched-pairs signed-rank test was used to investigate significant gene expression differences between tumor and normal tissue. For the statistical analysis of cell culture experiments a one-way ANOVA was used, and normally distributed data were presented as mean (SD). If the normality test failed, data were presented as median (IQR) and were calculated with a Kruskal–Wallis one-way analysis. The correlation between cancer cell lipid droplet area and *SREBF1* expression was performed by using Spearman correlation.

For the statistical analysis, we used the Sigma Plot 14 software package. A *p*-value of less than 0.05 was considered significant. Data were visualized with GraphPad Prism 9 (GraphPad Software, San Diego, CA, USA) and SPSS 26.0 (IBM, Ehningen, Germany).

## 3. Results

### 3.1. Missing a Clear Relationship between LD Formation, SREBF1 Expression, and Cell Sensitivity to Chemotherapeutics

Six different lung cancer cell lines (H1299, H1563, H1975, H1573, H1437, and H661) were cultured in BM for 18 h or 48 h and analyzed for LD formation and *SREBF1* expression. The highest number of LDs was found in H1573 cells although H1437 and H1299 cells also formed a lot of LDs. By contrast, H661 and H1975 cells had only minor quantities of LDs, whereas H1563 cells had no LDs (Figure 1A). As expected, cells with higher numbers of LDs also showed a higher expression of the *SREBF1* gene (Figure 1B) and the number of LDs positively correlated with *SREPF1* expression (Figure 1C). Further analyses of intracellular LDs and cell viability after treatment with cisplatin demonstrated that while cisplatin induced LD formation in H1563, H1975, and H661 cells, it decreased cell viability by about 50%. Likewise, cisplatin strongly reduced the viability of H1437 and H1299 cells even though these cells had large numbers of LDs (Appendix A and Figure 1D). Among all studied cell lines, only H1563 cells, which formed LDs in response to etoposide, were also resistant to this drug (Figure 1D). Notably, the viability of H1573 cells was lowest as compared to other cell lines (Figure 1D). This latter was related to the strong aggregation of H1573 cells, which required a harder trypsinization to dissociate these cells before for the flow cytometry analysis.

### 3.2. Morphological Characteristics of H1299 but Not H1563 Cells Change in Serum-Free Neutrophil Degranulation Medium (NDM) as Compared to Basal Medium (BM)

For the following experiments, we chose two adenocarcinoma cell lines: H1299, established from a lung lymph node metastasis, and H1563 established from a lung adenocarcinoma. H1299 cells abundantly formed LDs whereas H1563 did not show any traces of LDs (Figure 1). Both cell lines had typical morphology when cultured in serum-free BM up to 48 h (Figure 2). In NDM, already after 18 h, H1299 cells started to detach from the culture plates (Figure 2A). However, even after 48 h, these cells (attached and detached) preserved a similar viability as in BM (living cell% in BM vs. NDM, mean (SD): 80.4 ± 2.5 vs. 72.4 ± 3.7, n.s., *n* = 6 and *n* = 3 independent experiments, respectively). To confirm that detached floating cells remained alive, we cultured these cells in a basal growth medium. At 18 h, cells attached well to the cell culture plates, started to proliferate, and showed typical morphology. In contrast, H1563 cells did not change morphology and adhesion properties in NDM as compared to BM (Figure 2). Notably, changes in culture media did not affect LD formation, i.e., H1299 formed LDs whereas H1563 had no LDs. In both cell lines, the formation of LDs was not affected by the medium used (data not shown).

### 3.3. The Expression of Genes Related to Lipid Metabolism Differs between H1299 and H1563 Cells

We next performed RNA-seq on H1299 and H1563 cell lines grown in BM and NDM. In BM, H1299 and H1563 cells showed 13,545 differentially expressed genes (DEGs) belonging to 408 significant gene ontology (GO) terms characterizing the variation in gene expression levels across diverse human tissues (Figure 3A). The comparison between NDM and BM showed 883 DEGs in H1299 cells, of which 485 were up- and 389 downregulated. The gene set enrichment analysis (GSEA) revealed 43 significant GO terms associated with the DEGs. In H1563 cells, the number of DEGs between NDM and BM was 1731, of which 854 were up- and 877 downregulated, leading to 294 significant GO terms according to GSEA (Figure 3B). By using the DESeq2 Bioconductor package, we searched for the DEGs associated with NDM/BM differences between the cell lines. We detected 674 DEGs related to the differences in the medium effect between H1299 and H1563 cell lines, many of which outlined the regulation of sterol and cholesterol biosynthesis processes (Figure 3C,D).

As shown in Table 1, genes related to cholesterol synthesis process (GO:0045540), including *SREBF1*, *HMGCR, APOE,* and *HMGCS1,* were coherently lower expressed in H1563 than in H1299 cells.

We validated randomly selected genes of lipid metabolism with differential expression in H1299 and H1563 cells in BM or NDM by RT-qPCR. In line with RNA-seq data, the expression of *SREBF1*, *HMGCS1*, *HMGCR*, and *LDLR* was higher in H1299 than in H1563 cells. When compared to BM, *SREBF1* and *LDLR* were significantly lower in both cell lines in NDM whereas the expression of *HMGCS1* was reduced only in H1563. The expression of *HMGCR* was not affected by the medium (Figure 4).

### 3.4. Lipid Droplets (LDs)-Bearing H1299 Cells Are More Sensitive to Etoposide Than H1563 Cells Nonbearing LDs

In the following experiments, we assessed the effects of etoposide and cisplatin on H1299 and H1563 cells’ apoptosis and metabolic activity. As shown in Figure 5A,B, cisplatin significantly reduced the viability of both cell lines, independently of the cell culture medium. In both BM and NDM, H1563 cells were resistant to etoposide whereas the viability of H1299 cells declined by about 50% in response to etoposide. Concomitantly, cisplatin reduced the metabolic activity of both cell lines cultured in BM or NDM while the effect of etoposide was only significant in H1299 cells (Figure 5C,D).

### 3.5. Cisplatin and Etoposide Induce LD Formation in H1563 Cells but Do Not Affect LD in H1299 Cells

It is known that chemotherapies can interfere with LD formation [30], therefore we investigated LDs in cancer cells treated with etoposide or cisplatin for 48 h As illustrated in Figure 6A,C, H1299, but not H1563, cells formed LDs in BM. Treatments with cisplatin or etoposide had no effect on LD formation in H1299, however, they induced LDs in H1563 cells (Figure 6C). The same scenario was observed when cells were cultured in NDM (data not shown). The quantification of the LDs did not show significant differences between H1299 treated or not treated (Figure 6A,B), while in H1563 cells, the drugs induced LDs (Figure 6C,D).

As expected, higher levels of SREBF1 mRNA and protein were found in H1299 cells forming LDs than in H1563 cells without LDs (Figure 7A,B). Although chemotherapeutics reduced *SREBF1* gene expression in both cell lines, this reduction did not parallel with changes in LDs, i.e., H1299 did not decrease LD numbers whereas H1563 began to form LDs. According to dot blots, drugs did not induce remarkable changes in cellular SREBF1 protein levels.

The absence of a direct correlation between *SREBF1* and LDs in cells treated with cisplatin and etoposide prompted us to examine the effects of these drugs on other genes, which are related to different pathways that link cell viability and LDs formation. From our RNA-seq data set, we selected the following genes associated with lipid metabolism (*AGPAT2*), apoptosis (*SERPINE1* and *THBS1*), DNA repair processes (*CDKN1A* and *TOPBP1*), and oncogenesis (*WNT3*). As shown in Figure 7, at a baseline, H1299 cells had significantly higher expression of *AGPAT2* and *SERPINE1* but a lower expression of *WNT3* than H1653 cells. The expression of *SERPINE1* decreased significantly in H1299 cells after treatment with cisplatin or etoposide, while *AGPAT2*, *CDKN1A, TOPBP1, THBS1,* and *WNT3* expressions remained unaffected. In contrast to H1299, in cisplatin- and etoposide-treated H1563 cells, the expression of *CDKN1A* significantly increased while *WNT3* decreased. Only in response to cisplatin did the expression of *SERPINE1* significantly increase and *AGPAT2* decrease in H1653 cells (Figure 7C,D). Furthermore, H1563 cells treated with cisplatin or etoposide showed a slight decrease in *THBS1* expression, and no change in *TOPBP1* expression (Figure 7G). Hence, the chemotherapeutic-induced changes in expression patterns of the analyzed genes implied that the viability and LDs of H1653 and H1299 cells might rely on several pathways not necessary involving *SREBF1*.

### 3.6. The Lower SREBF1 Expression in Tumor Than in Nontumor Tissues from Patients with Lung Adenocarcinoma and Squamous Cell Carcinoma Correlates with a Better Overall Survival

Based on our findings that *SREBF1* expression was affected by the cell culture conditions and chemotherapeutics, and on previous reports that a high expression of *SREBF1* correlated with the progression of colon and pancreatic cancers [31,32], we aimed to investigate a potential relationship between *SREBF1* expression and the survival of patients suffering from lung cancer. For this, 292 patient samples (tumor and corresponding normal lung tissues from the same patient) were analyzed for the *SREBF1* expression (Figure 8). We found a significant downregulation of *SREBF1* in adenocarcinoma (ADC) and although statistically nonsignificant, also a downregulation in squamous cell carcinoma (SQCC) (Figure 8A, Appendix A). In general, *SREBF1* expression was more spread over a wide area in tumor samples compared to normal lung tissues (Appendix A). Interestingly, the downregulation of *SREBF1* was significant in early stage I of ADC (Figure 8B). Using Cutoff Finder [29] and univariate Cox regression analyses, we observed an increased overall survival (OS) hazard ratio for patients with ADC (HR 1.675 (95% CI 1.2–2.751)) or SQCC (HR 2.305 (95% CI 1.15–4.619) with lower *SREBF1* levels (Appendix A). Interestingly, we observed a stronger influence of *SREBF1* on the OS of patients with SQCC (*p* = 0.015) compared to ADC (*p* = 0.039) (Figure 8C). Multivariate cox-regression analyses revealed that *SREBF1* (*p* = 0.012), age (*p* = 0.007), and gender (*p* = 0.025) were prognostic factors for OS in patients with SQCC while *SREBF1* failed as a prognostic factor in a multivariate analysis of patients with ADC. To investigate the effect of *SREBF1* on survival in the different stages of the disease, we used the same cutoffs and analyzed patient survival data regarding the pathological stage. Here, we observed that a lower *SREBF1* expression correlated with a better survival in ADC stage I, but not in stages II and III (Figure 8D). There were no significant or reliable results for SQCC since either the *p*-value was not significant, or the groups were too small (Appendix A).

## 4. Discussion

In this study, we explored the relationship between lipid droplet (LD) formation, *SREBF1* expression, and lung cancer cell sensitivity to chemotherapeutics, such as cisplatin and etoposide. We also investigated the impact of neutrophils on this relationship. According to clinical and preclinical studies, the accumulation of neutrophils in tumors and high counts of peripheral blood neutrophils are related to a poor prognosis of patients [33]. As neutrophils respond to various inflammatory and cancerous signals, they are among the most abundant immune cells in the TME influencing tumor development in different ways. There is also evidence for neutrophil degranulation in the TME, which results in the release of various cytokines, chemokines, proteases, and free radicals [34]. These latter, dependent on their quantitative and qualitative properties, may influence tumor development in different ways. Recent data provide novel evidence that neutrophils can release vesicles, which can provide stored lipids necessary for cancer cells survival [10,35,36,37,38]. Cancer cells are known to take up lipids and simultaneously enhance lipid synthesis, which can lead to the accumulation of LDs [39]. During cancer cell proliferation, LDs can provide an energy reservoir and components necessary for cell membranes, thereby maintaining cell survival [40].

Therefore, we employed an experimental model, in which the LD status and viability of lung adenocarcinoma cells were studied not only in a serum-free basal medium (BM) but also in medium containing all soluble factors released upon neutrophil degranulation (NDM). It is important to point out that the degranulation of freshly isolated neutrophils was induced by ultrasound without adding external substances to avoid any specific degranulation response. For example, various pathogens or chemical substances can induce neutrophil degranulation via the engagement of specific neutrophil surface receptors and granules, and therefore the degree of neutrophil degranulation may vary [41], which would require separate experimentations.

Cancer cells’ abilities to adapt to the environment are of critical importance for their survival. Lipid mobilization and LD accumulation are among the mechanisms involved in this process of adaptation [42,43]. The sterol regulatory element-binding proteins (SREBPs) are key regulators of synthesis and trafficking of cholesterol and other lipids and are implicated in maintaining lipid homeostasis and LD biogenesis [5,44]. Indeed, our initial experiments with different lung cancer cell lines confirmed that higher expression levels of *SREBF1* were related to a more pronounced LD formation. It was not surprising that a transcriptome analysis of H1563 (without LDs) and H1299 (with LDs) cells cultured in BM revealed many DEGs related to sterol metabolism and cholesterol biosynthesis processes including *SREBFs*. When comparing the transcriptome profiles of cells cultured in NDM against those in BM, we again observed a significantly lower expression of many genes involved in the regulation of sterol and cholesterol biosynthesis in H1563 versus H1299. Even though the expression of many genes related to lipid pathways were altered in NDM, the formation of LDs was not affected, and NDM had no significant effect on cell viability. It is known that LD biogenesis in cancer can also be controlled by SREBP1-independent mechanisms [3].

It is of interest to point out that H1299 cells, but not H1563 cells, when cultured in NDM, to a large extent lost their adhesion property, i.e., most of the H1299 cells were detached from the culture plates but preserved their viability and proliferative properties. In vivo, this may facilitate the neutrophil degranulate-mediated transport of lung cancer cells into different tissues. These findings are of potential interest to pursue further, considering that tumor-associated neutrophils, lipid-metabolism-related genes, such as *SREBF1*, and LD formation are hallmarks contributing to the cancer cell survival and drug resistance [45,46,47].

In our experiments, when H1563 and H1299 cell cultures in serum-free BM or NDM were treated with cisplatin or etoposide, we found no direct linkage between LD formation and cancer cell sensitivity to a specific chemotherapeutic. For example, under starving conditions, H1563 cells started to form LDs in response to cisplatin and etoposide but were resistant only against etoposide. We initially hypothesized that LD formation in H1563 cells and their low sensitivity to etoposide may depend on increased *SREBF1* expression. However, independently of the media or drugs used, *SREBF1* expression decreased in H1563, excluding a role of *SREBF1* in this scenario. In general, the mechanisms underlying LD biogenesis and formation remain not well understood. One cannot exclude that LD’s size, composition, and localization rather than the LD area will determine their role in anticancer drug resistance [48].

Alpsoy et al. claimed that lower levels of *TOPBP1* (DNA topoisomerase II binding protein (1)) might determine cancer cell resistance against etoposide [49]. Moreover, *TOPBP1* overexpression was associated with genome instability in H1299 cells [50]. To withstand chemotherapy, cancer cells can activate cyclin-dependent kinase inhibitor 1A (*CDKN1A*, p21), a regulator of cell cycle and repair of DNA damage suggesting that *CDKN1A* overexpression can help cells to escape apoptosis [51]. In fact, H1563 had a much higher basal expression of *CDKN1A* than H1299 cells, which further increased in response to etoposide. The expression of *AGPAT2*, an enzyme involved in the glycerophospholipid/triacylglycerol biosynthesis pathway, has been suggested to promote survival and etoposide resistance of cancer cells, and to be directly involved in LD formation [52]. In line with these results, data from our gene expression analysis revealed that *TOPBP1* expression was lower while *AGPAT2* was higher in H1563 than in H1299 cells. Thus, a lower expression of *TOPBP1* and a high expression of *AGPAT2* and *CDKN1A* in H1563 cells may at least partially explain resistance to etoposide. However, why these cells are sensitive to cisplatin remains unclear, although a recent study reported that *CDKN1A* upregulation reduced cancer cell sensitivity to cisplatin [53]. *SERPINE1*, also known as *PAI-1*, is associated with cellular senescence and poor cancer prognosis [54]. According to Pavon et al. [55], a higher *SERPINE1* expression in cancer leads to a lower sensitivity to cisplatin. Indeed, basal levels of *SERPINE1* expression were much lower in H1563 than in H1299 cells, which might provide one putative explanation why H1563 cells are sensitive to cisplatin. However, this cannot explain why H1299 cells having a higher expression of *SERPINE1* are as sensitive to cisplatin as H1563 cells. One can also mention that if compared to H1299 cells, H1563 cells have a higher baseline expression of *AGPAT2* (1-Acylglycerol-3-Phosphate O-Acyltransferase (2)), *THBS1* (Thrombospondin 1), and *WNT3* (Wnt Family Member (3)). The elevated expression of *THBS1* and *WNT3* genes confers cancer cell resistance to therapies through diverse mechanisms [56,57]. Hence, these genes can contribute to H1563 resistance to etoposide as well. Taken together our data in vitro showed that changes in the composition of microenvironment strongly affected the cancer cell transcriptome, and that changes in the expression of single genes, such as *SREBF1*, were not sufficient to explain the dynamics of LD formation and cell sensitivities to specific chemotherapies.

As mentioned earlier, the *SREBF1* gene was one of the highlighted genes linking oncogenic signaling with lipid metabolism, and an elevated *SREBF1* expression was correlated with cancer progression and metastasis of several cancer types, including lung cancer [58]. Therefore, we checked for *SREBF1* expression in tumor and adjacent nontumor tissues from lung cancer patients. Indeed, a lower *SREBF1* expression in tumors correlated with a better overall patient survival. Our findings in the lung cancer patient cohort were in concordance with other studies performed in patients with other cancer types [31,32,59]. When comparing the expression of *SREBF1* in pathological stages, we observed a lower expression in early stages and a higher expression in later stages (stage II and III). Especially in patients with early stages of adenocarcinoma, a higher expression of *SREBF1* resulted in a worse prognosis. These data might seem contradictory but might also give a hint that changes in *SREBF1* expression and LD formation in cancer cells might be an essential early step in tumorigenesis. A lower inflammatory state at early stages of ADC might be another possible explanation for our findings [60,61]. The *SREBF1* value as a biomarker might be lost during later stages of lung cancer, which are characterized by a persistent and advanced inflammatory state.

## 5. Conclusions

A direct relationship between *SREBF1* expression and LD formation can be found in lung cancer cell lines cultured in serum-free BM. However, this relationship seems to be lost when cells are cultured in serum-free NDM and/or treated with chemotherapeutics, such as etoposide or cisplatin. Despite a reduction in *SREPF1* expression, some lung cancer cells may exhibit LD formation in response to chemotherapeutics and acquire resistance. Likewise, we provided experimental evidence that two cell lines were similarly sensitive to cisplatin even though H1299 cells preserved LDs and H1563 cells began to form LDs. In response to etoposide, H1563 cells also formed LDs but were etoposide-resistant, while H1299 cells, which preserved LDs, were as sensitive to etoposide as to cisplatin. According to previous reports, the levels of LDs can determine cancer cell sensitivity against cisplatin [62,63]; however, our data were not congruent with these suggestions. Moreover, in our cell model, components of activated neutrophils did not influence LD formation and cell sensitivities to cisplatin and etoposide. Hence, a more detailed analysis of the microenvironment and LDs is required to understand if and how LDs can help cancer cells to resist chemotherapies.

In general, the current data support the notion that LDs and *SREBF1* have strong protumor functions in various cancers. In a cohort of 292 patients with non-small-cell lung cancer, a lower *SREBF1* expression in tumors than in adjacent nontumor tissue correlated with overall better survival, specifically in patients with adenocarcinoma at stage I. Thus, *SREBF1* expression can be considered as a prognostic factor for better patient overall survival. The level of *SREBF1* is tightly controlled by endogenous sterol levels via negative feedback regulation. Therefore, multidimensional approaches are needed to explore all the smallest parts of the interplay between *SREBF1* expression, lipid metabolism, and LD formation in cancer cells.

## Figures and Tables

**Figure 1 cancers-14-04454-f001:**
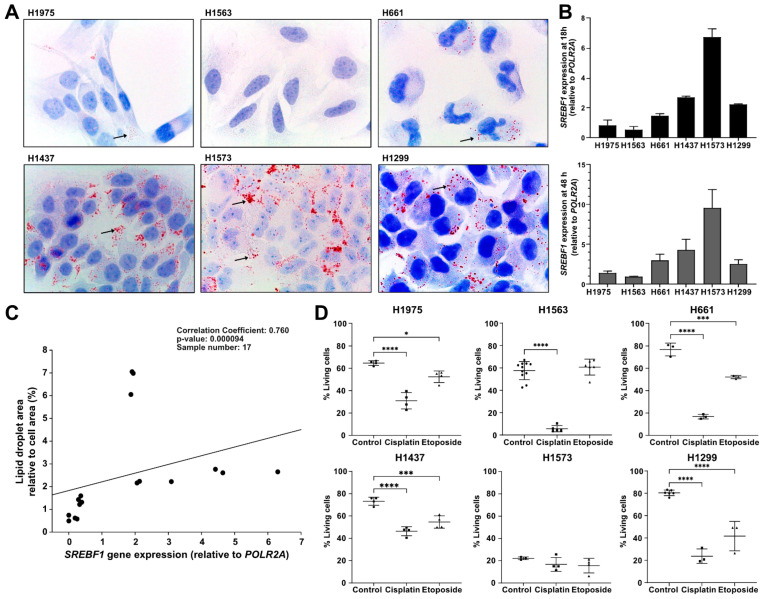
Lipid droplet formation, SREBF1 gene expression, and viability assay in six lung cancer cell lines. (**A**) Lipid droplets were stained with Oil red O and nuclear components with hematoxylin after cell culture in BM for 18 h. Arrows indicate lipid droplets. Images were taken at 1000-fold magnification using a 100× oil immersion objective (Leica). One representative experiment out of three independent experiments is shown. (**B**) Six cell lines were incubated in BM for 18 h or 48 h before RNA isolation and analysis for *SREBF1* expression. Gene expression was carried out in duplicate from three independent experiments and data are presented as mean (SD). (**C**) The correlation between the LDs (**A**) and *SREBF1* expression (**B**) was performed by using Spearman’s correlation, *n* = 17 resulting from two or three independent experiments per cell line from a total of six cell lines. (**D**) Cells were cultured in BM for 48 h with or without 20 µg/mL cisplatin or 100 µM etoposide. The cell viability was assessed by Annexin V 7AAD Apoptosis Detection Kit. Living cells (%) were determined against nontreated controls for each medium separately. Number of independent experiments: *n* = 3 for H661, H1299 etoposide and cisplatin, H1573 control; *n* = 4 for H1975, H1437; *n* = 5 for H1563 cisplatin; *n* = 6 for H1299 control, H1563 etoposide; *n* = 11 for H1563 control. *p*-values of treatment vs. control cells were calculated with a one-way ANOVA. Data are presented as mean (SD). A *p*-value below 0.05 was considered as significant: * *p* < 0.05, *** *p* < 0.001, and **** *p* < 0.0001. BM, basal medium.

**Figure 2 cancers-14-04454-f002:**
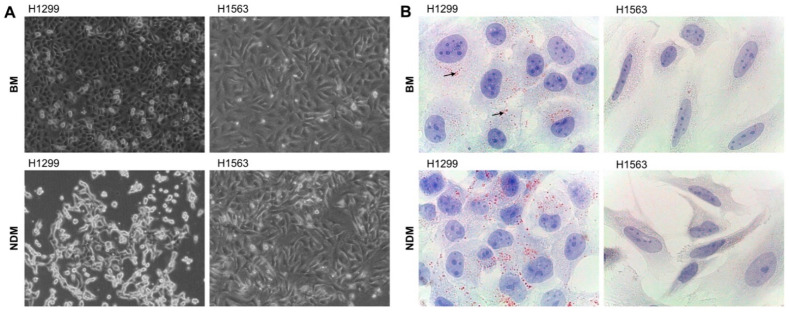
Morphology, and lipid droplet (LD) analysis of H1299 and H1563 cells cultured in NDM vs. BM: (**A**) Cell morphology and adhesion to the plate after 18 h incubation in BM (top images) or NDM (bottom images). Images were taken with a Leica DM IL LED microscope (Leica) equipped with a FLEXACAM C1 camera (Leica) at 100-fold magnification. (**B**) Cells were stained for lipid droplets with Oil red O and for nuclear components with hematoxylin after culture for 18 h in BM (top) or in NDM (bottom). Arrows indicate lipid droplets. Images were taken at 1000-fold magnification using 100× oil immersion objective (Leica). One representative experiment out of four independent experiments is shown. BM, basal medium; NDM, neutrophil degranulation medium.

**Figure 3 cancers-14-04454-f003:**
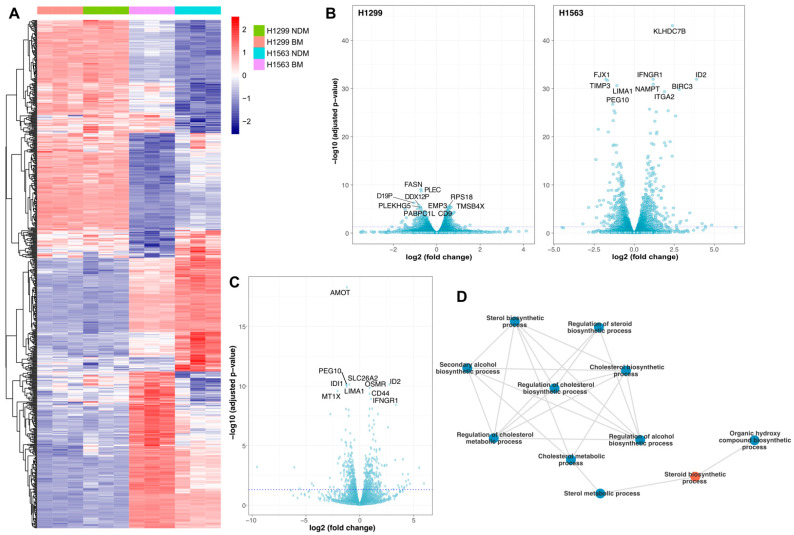
Transcriptome analysis of H1299 and H1563 cells by RNA-sequencing. (**A**) The heatmap shows the normalized read counts for the differentially expressed genes across the four conditions (H1299 in NDM, H1299 in BM, H1563 in NDM, and H1563 in BM) calculated by the interaction term. Data were normalized row-wise to make the expression levels comparable across different genes (color of the heatmap: blue, low expression; red, high expression). The colors above the heatmap indicate cell lines cultured in different media. (**B**) Volcano plots show the differential expression results of the medium effect (in other words, the differences between NDM and BM) in H1299 cells (left) and H1963 cells (right). The *x*-axis represents the log2 (fold change) values and the *y*-axis represents the −log10 transformed adjusted *p*-values. Ten genes with the highest significance (namely, with the lowest adjusted *p*-values) are labeled. (**C**) This differential expression result reflects how the medium effect differs across H1299 and H1563 cell lines. The volcano plot shows effect-size as log2 fold change of the cell-line-dependent medium effect on the *x*-axis, and the significance of those differences on the *y*-axis as −log10 (*p*-values). Ten genes with the highest significance (namely, with the lowest adjusted *p*-values) are labeled. (**D**) A network plot describes interactions of the 64 most significant GO terms based on the DEGs. The result list of GO terms was trimmed to 50% of the original length by REVIGO based on the semantic similarities. The largest cluster of GO terms belongs to sterol biosynthesis process (red node). The width of the links between the nodes represent the functional similarities calculated by the genetic overlap between nodes. BM, basal medium; NDM, neutrophil degranulation medium.

**Figure 4 cancers-14-04454-f004:**
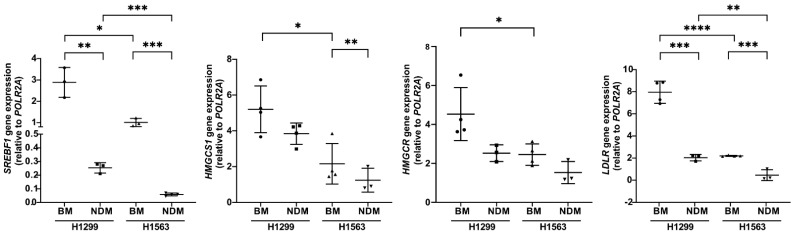
Expression of lipid-related genes in H1299 and H1563 cells cultured in BM and NDM. Cells were incubated in BM or in NDM for 18 h and RNA was isolated and analyzed by real-time qPCR. Data are shown as mean (SD) from *n* = 3 independent experiments except for *SREBF1* H1563 NDM, *LDLR* H1299 and H1563 BM, *HMGCS1* H1299 BM/NDM and H1563 BM, *HMGCR* H1299 and H1563 BM with *n* = 4. All analyses were carried out in duplicate. *p*-values were calculated with a one-way ANOVA. A *p*-value below 0.05 was considered as significant: * *p* < 0.05, ** *p* < 0.01, *** *p* < 0.001, and **** *p* < 0.0001. BM, basal medium; NDM, neutrophil degranulation medium.

**Figure 5 cancers-14-04454-f005:**
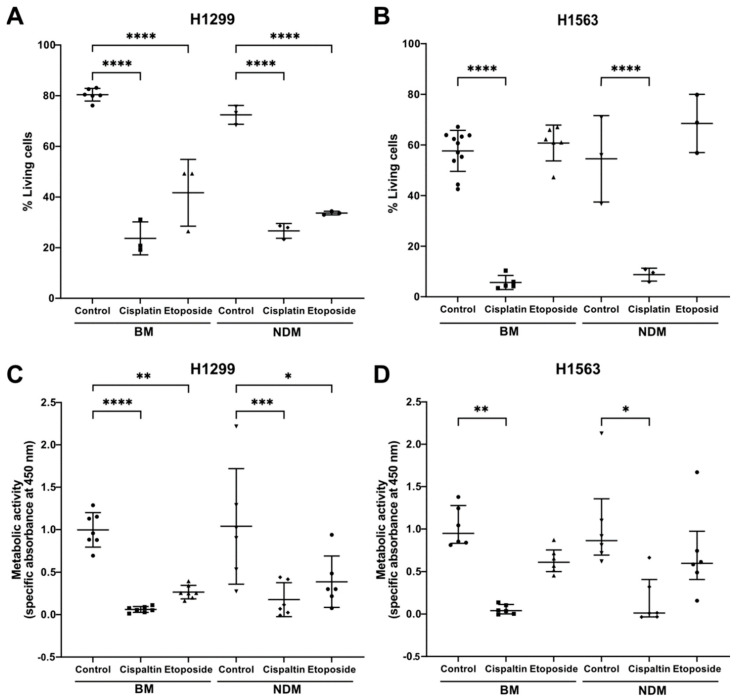
Effects of cisplatin and etoposide on cell viability and metabolic activity in BM and NDM. Cells were cultured for 48 h in BM or NDM in presence or absence of 20 μg/mL cisplatin or 100 µM etoposide. Cell viability analysis by Annexin V 7AAD Apoptosis Detection Kit for H1299 cells (**A**) or H1563 cells (**B**). Livings cells (%) were determined against nontreated controls for each medium separately. *p*-values of treatment vs. control were calculated with a one-way ANOVA. Data are presented as mean (SD). A *p*-value below 0.05 was considered as significant. **** *p* < 0.0001. *n* = 3 for cell viability (except H1563 BM: *n* = 5). (**C**,**D**). Cell metabolic activity assay for H1299 (**C**) and H1563 (**D**) cells. (**C**) Data are presented as mean (SD) and *p*-values calculated with a one-way-ANOVA and (**D**) data presented as median (IQR) and *p*-values calculated with a Kruskal–Wallis test. A *p*-value below 0.05 was considered significant. * *p* < 0.05, ** *p* < 0.01, *** *p* < 0.001, **** *p* < 0.0001, *n* = 6; BM, basal medium; NDM, neutrophil degranulation medium.

**Figure 6 cancers-14-04454-f006:**
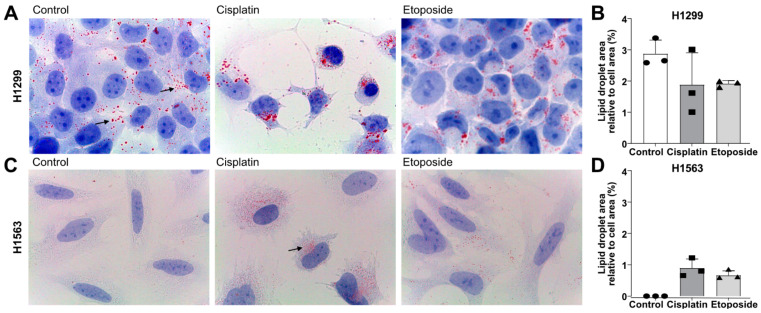
Influence of cisplatin and etoposide treatments on lipid droplet (LD) formation. LDs in H1299 (**A**) and H1563 (**C**) were stained with Oil red O after 48 h incubation in BM (basal medium) in presence or absence of 20 μg/mL cisplatin or 100 µM etoposide. Images are representative out of four independent experiments. Arrows indicate LDs. Images were taken at 1000-fold magnification using a 100× oil immersion objective (Leica). The quantification of LDs was based on the percentage of LD area relative to cell area of *n* = 3 independent experiments for H1299 (**B**) or H1563 (**D**). Circles represent control cells incubated with BM, squares cisplatin-treated cells and triangles etoposide-treated cells.

**Figure 7 cancers-14-04454-f007:**
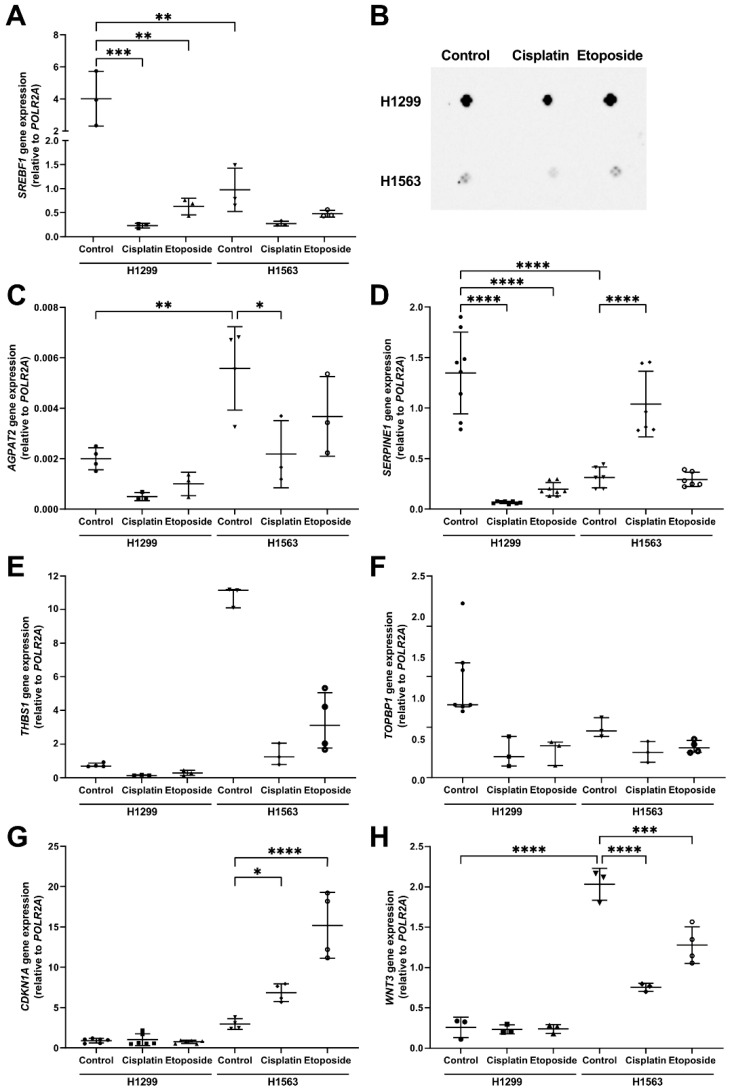
Transcript and protein levels of SREBF1, and expression of other genes related to lipid metabolism, apoptosis, DNA-repair, and oncogenesis processes. Cells were incubated for 48 h in the presence or absence of 100 μM etoposide or 20 μg/mL cisplatin in serum-free basal medium (BM). RNA was isolated and analyzed by real-time qPCR or cells were lysed, and protein levels were assayed. (**A**) *SREBF1* gene expression; (**B**) SREBF1 protein levels analyzed by dot blots (22 μg protein per dot); (**C**) *AGPAT2*; (**D**) *SERPINE1*; (**E**) *THBS1*; (**F**) *TOPBP1*; (**G**) *CDKN1A*; and (**H**) *WNT3* gene expression. Gene expression was carried out in duplicate from three to eight independent experiments. The mean (SD) was calculated with a one-way ANOVA for *SREBF1*, *AGPAT2, SERPINE1,* and *CDKN1A* genes, whereas *THBS1, WNT3*, and *TOPBP1* were analyzed with a Kruskal–Wallis test and presented as median (IQR). A *p*-value below 0.05 was considered as significant: * *p* < 0.05, ** *p* < 0.01, *** *p* < 0.001, and **** *p* < 0.0001.

**Figure 8 cancers-14-04454-f008:**
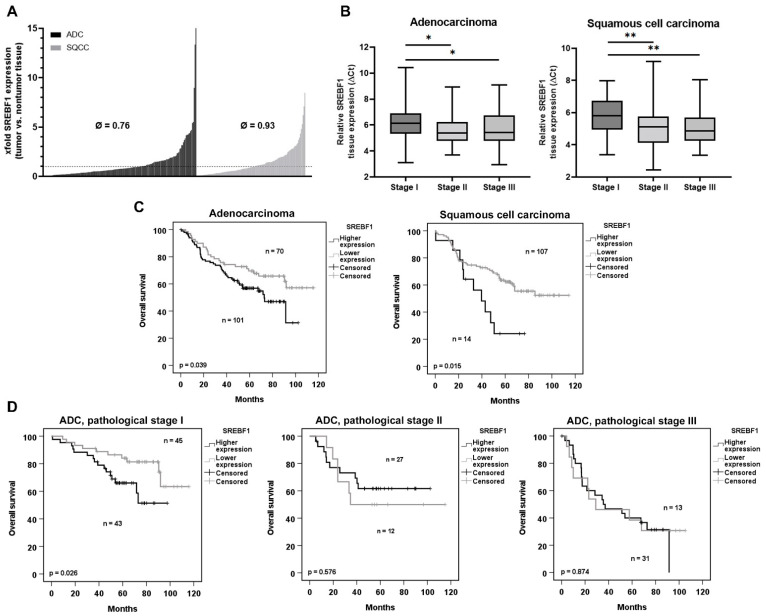
*SREBF1* expression in NSCLC patients. (**A**) Fold change (∆∆Ct) was calculated using relative gene expression of *SREBF1* in tumor compared to normal tissue in 292 patients. (**B**) Relative *SREBF1* expression in adenocarcinoma and squamous cell carcinoma. Gene expression was normalized with two housekeeping genes (*ESD* and *RPS18*). Please note that higher values mean a lower gene expression. (**C**,**D**) Kaplan–Meier survival curves of *SREBF1* tumor expression. Cutoffs for the separation of the groups in (**C**), (**D**) were calculated using the R tool Cutoff Finder [29]. ADC = adenocarcinoma, SQCC = squamous cell carcinoma, Ø = median, * *p* < 0.05, ** *p* < 0.005.

**Table 1 cancers-14-04454-t001:** Regulation of cholesterol biosynthesis process (GO:0045540) in H1563 vs. H1299 in BM.

Gene Name	Base Mean	Log2 Fold Change	LfcSE	Stat	*p*-Value	Padj	Direction
*FGF1*	36.33	−6.67	1.08	−6.17	6.71 × 10^−10^	1.68 × 10^−10^	▼
*ABCG1*	937.21	6.65	0.22	30.56	4.10 × 10^−205^	4.10 × 10^−204^	▲
*MVD*	4164.03	−3.51	0.14	−25.53	9.40 × 10^−144^	4.50 × 10^−143^	▼
*APOE*	368.96	−3.23	0.24	−13.69	1.16 × 10^−42^	9.48 × 10^−43^	▼
*FASN*	26,278.63	−3.08	0.15	−20.98	9.99 × 10^−98^	2.49 × 10^−97^	▼
*SREBF1*	6809.99	−2.99	0.11	−26.62	4.10 × 10^−156^	2.20 × 10^−155^	▼
*FDPS*	14,122.46	−2.94	0.09	−32.41	2.20 × 10^−230^	2.70 × 10^−229^	▼
*SCD*	105,179.20	−2.90	0.13	−22.52	2.40 × 10^−112^	7.50 × 10^−112^	▼
*LSS*	7094.47	−2.67	0.12	−22.58	6.60 × 10^−113^	2.10 × 10^−112^	▼
*FDFT1*	9258.84	−2.44	0.10	−25.10	4.60 × 10^−139^	2.10 × 10^−138^	▼
*MVK*	836.61	−1.92	0.12	−15.87	1.02 × 10^−56^	1.19 × 10^−56^	▼
*IDI1*	7365.73	−1.74	0.12	−14.88	4.35 × 10^−50^	4.31 × 10^−50^	▼
*HMGCS1*	11,484.75	−1.63	0.11	−15.14	8.61 × 10^−52^	8.88 × 10^−52^	▼
*TM7SF2*	754.88	−1.53	0.17	−9.05	1.48 × 10^−19^	5.72 × 10^−20^	▼
*NFYB*	1985.44	1.38	0.11	13.10	3.26 × 10^−39^	2.44 × 10^−39^	▲
*HMGCR*	8898.55	−1.26	0.08	−15.10	1.57 × 10^−51^	1.60 × 10^−51^	▼
*DHCR7*	3593.68	−1.16	0.10	−11.66	2.03 × 10^−31^	1.20 × 10^−31^	▼
*ELOVL6*	3212.98	1.15	0.10	11.23	2.85 × 10^−29^	1.58 × 10^−29^	▲
*RAN*	11,606.11	−1.08	0.12	−9.29	1.49 × 10^−20^	6.01 × 10^−21^	▼
*ERLIN2*	2056.03	0.96	0.10	9.35	8.39 × 10^−21^	3.43 × 10^−21^	▲
*MBTPS2*	2285.09	0.92	0.10	9.14	6.18 × 10^−20^	2.43 × 10^−20^	▲
*CYP51A1*	5903.00	−0.92	0.09	−9.68	3.80 × 10^−22^	1.64 × 10^−22^	▼
*SEC14L2*	773.96	0.81	0.11	7.62	2.44 × 10^−14^	7.60 × 10^−15^	▲
*SQLE*	7182.00	−0.79	0.10	−7.92	2.30 × 10^−15^	7.49 × 10^−16^	▼
*GPAM*	580.32	−0.77	0.13	−5.95	2.67 × 10^−9^	6.48 × 10^−10^	▼
*SREBF2*	6029.96	−0.65	0.09	−7.29	3.10 × 10^−13^	9.17 × 10^−14^	▼
*SCAP*	2424.48	0.62	0.11	5.81	6.10 × 10^−9^	1.46 × 10^−9^	▲
*SC5D*	3318.08	0.60	0.10	6.27	3.73 × 10^−10^	9.45 × 10^−11^	▲
*PRKAA1*	4033.46	−0.59	0.11	−5.30	1.16 × 10^−7^	2.60 × 10^−8^	▼
*KPNB1*	26,881.55	−0.57	0.09	−6.04	1.51 × 10^−9^	3.73 × 10^−10^	▼
*SP1*	5089.01	−0.56	0.09	−6.35	2.13 × 10^−10^	5.48 × 10^−11^	▼
*GGPS1*	1151.02	0.54	0.11	4.75	2.07 × 10^−6^	4.35 × 10^−7^	▲
*ERLIN1*	1600.73	−0.51	0.12	−4.32	1.59 × 10^−5^	3.17 × 10^−6^	▼
*NFYA*	1415.55	−0.41	0.11	−3.77	1.66 × 10^−4^	3.10 × 10^−5^	▼
*ACACA*	5645.03	0.41	0.12	3.49	4.78 × 10^−4^	8.65 × 10^−5^	▲
*SOD1*	8249.72	0.37	0.12	3.14	1.71 × 10^−3^	2.97 × 10^−4^	▲
*NFYC*	2042.66	−0.29	0.09	−3.12	1.82 × 10^−3^	3.16 × 10^−4^	▼
*MBTPS1*	6905.10	−0.25	0.08	−3.04	2.37 × 10^−3^	4.07 × 10^−4^	▼
*ACACB*	578.19	0.23	0.13	1.80	7.18 × 10^−2^	1.07 × 10^−2^	▲

Arrows indicate up-(red) or downregulated (blue) genes in H1563 compared to H1299 cells.

## Data Availability

All datasets generated and analyzed during the current study are available from the corresponding author on reasonable request.

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
