# Peer review of "Lung Adenocarcinoma Cell Sensitivity to Chemotherapies: A Spotlight on Lipid Droplets and SREBF1 Gene"

_cancers, 2022, doi:10.3390/cancers14184454_

Round 1

Reviewer 1 Report (Previous Reviewer 1)

The study entitled „Interplay between microenvironment and lung adenocarcinoma cell sensitivity to therapies: a spotlight on SREBF1 gene” is a second, much improved submission of the manuscript.

The authors extended number of cell lines used in in vitro experiments up to six. Improved description of the methods and general flow of the manuscript.

However, there are some issues that need to be addressed:

The statement: “Remarkably, in the cell lines forming less or no LDs under BM conditions (H1563, H1975, and H661), the treatment with cisplatin resulted in a LD formation and in a strong reduction (by about 50%) of cell viability” does the results from observation only? Data not shown?

 In figure 2A – cell detachment, were the cells counted anyhow?

The description of Figure 4 should be improved. It is much more than just “the expression of SREBF1, HMGCS1, HMGCR, and LDLR were (should be “was”) higher in H1299 than in H1563 cells”

…and similarly in Figure 5 the authors do not address the effect of the NDM medium on cellular activity

What is the conclusion for the observations presented in Figure 7? Something is wrong with the labelling of Figure 7B…

In discussion, where is this conclusion drawn from? “In fact, a subset of H1563 cells surviving the treatment with cisplatin showed LD formation, which might be a reason why these cells survived [37]”

Unfortunately the manuscript remains vague in some moments. I.e. the rationale for application of NDM is not provided, nor the results of the application are discussed.

Author Response

The study entitled „Interplay between microenvironment and lung adenocarcinoma cell sensitivity to therapies: a spotlight on SREBF1 gene” is a second, much improved submission of the manuscript.

The authors extended number of cell lines used in in vitro experiments up to six. Improved description of the methods and general flow of the manuscript.

However, there are some issues that need to be addressed:

-       The statement: “Remarkably, in the cell lines forming less or no LDs under BM conditions (H1563, H1975, and H661), the treatment with cisplatin resulted in a LD formation and in a strong reduction (by about 50%) of cell viability” does the results from observation only? Data not shown?

Answer: This is correct, the data was not shown. Now we include it as supplementary figure S3.

-       In figure 2A – cell detachment, were the cells counted anyhow?

Answer: Unfortunately, we did not count the cells. To give an estimation: roughly about 2 thirds of the cells detached during overnight culture in NDM. However, we analyzed all (attached and detached) cells by FACS or metabolic assay and we didn’t find a significant difference in the cell concentration, viability, or metabolic activity of cells cultured in BM or NDM (figure 5). We also re-cultured detached cells in growth medium and they attached again to the culture vessels, showed typical morphology and started proliferation at 18 hours. This leads us to the suggestion that detached cells remain alive. We updated on this matter result and discussion sections (page 7, lines 303 and page 17, line 550).

-       The description of Figure 4 should be improved. It is much more than just “the expression of SREBF1, HMGCS1, HMGCR, and LDLR were (should be “was”) higher in H1299 than in H1563 cells”

Answer: Thank you very much for this comment. We added additional sentences to complete the description of figure 4 to the text. See page 10 lines 367 and following.

-       …and similarly in Figure 5 the authors do not address the effect of the NDM medium on cellular activity.

Answer: We included a phrase that the metabolic activity was not affected by the medium. Please see page 11, line 386

-       What is the conclusion for the observations presented in Figure 7? Something is wrong with the labelling of Figure 7B…

Answer: Thank you for this comment. We corrected the labelling in figure 7B and completely reworked the result part based on figure 7. We added reasons why we selected the presented genes and conclusions (see page 13). 

-       In discussion, where is this conclusion drawn from? “In fact, a subset of H1563 cells surviving the treatment with cisplatin showed LD formation, which might be a reason why these cells survived [37]”

Answer: We removed this conclusion as irrelevant.

-       Unfortunately the manuscript remains vague in some moments. I.e. the rationale for application of NDM is not provided, nor the results of the application are discussed. 

Answer: Thanks a lot for this comment. We reworked the Discussion and Conclusions parts in order to make our manuscript clear for the readers (see changes marked in red).

Reviewer 2 Report (Previous Reviewer 2)

1. The title of the paper is vague. Microenvironnmet includes many factor in context to cancer cells. Here authors are specifically discussing NDM media on lung cancer cells, LD formation capability and response to chemotherapeutics. Please be specific on title about what the manuscript is about. 

2. Co-relation analysis is required in LF formation capability and SREBF1 expression.

3. "Noticeably, the viability of H1573 268 cells in BM was low due to the difficulties in removing these strongly attached cells from 269 the culture plates." This is not to understand, authors could have used 0.25% trypsin or allowed longer time for trypsinization. 

4. Figure 2B and Figure 6 - Quantitation required

5. "However, in NDM already after 18 h H1299 cells got roundish and 293 started to detach from the culture plates (Figures 2A and B)." Having observe this phenotype I do not think cell viability analysis of drugs in NDM media and comparing the results of cell viability analysis in BM media is a good approach. For this reason, results section 2.4 is questionable. 

6. RNAseq analysis need more clarification. In Figure 3C, why the results were combined for both cell lines? 

7. Lines 407-412: The results presented here is diverting the authors from main messages of the manuscript.

Author Response

  1. The title of the paper is vague. Microenvironment includes many factor in context to cancer cells. Here authors are specifically discussing NDM media on lung cancer cells, LD formation capability and response to chemotherapeutics. Please be specific on title about what the manuscript is about. 

Answer: We changed the title to “Lung adenocarcinoma cell sensitivity to chemotherapies: a spotlight on lipid droplets and SREBF1 gene” to make it more specific to reflect aim of the manuscript.

  1. Co-relation analysis is required in LF formation capability and SREBF1 expression.

Answer: Thank you very much for this comment. We correlated LD area to SREBF1 gene expression and found a positive correlation between both parameters (see-updated figure 1). 

  1. "Noticeably, the viability of H1573 268 cells in BM was low due to the difficulties in removing these strongly attached cells from 269 the culture plates." This is not to understand, authors could have used 0.25% trypsin or allowed longer time for trypsinization. 

Answer: You are right. However, H1573 cells form cell aggregates. For FACS measurement, it is crucial to break these aggregates and to get single cells. This was only possible with increased trypsinization times, which unfortunately leads to reduced numbers of living cells. See the correction on page 6, line 280.

  1. Figure 2B and Figure 6 - Quantitation required

Answer: We updated figure 6 accordingly. For figure 2A, we found that the LD area was not affected by the medium used (data not shown). See page 8, line 305)

  1. "However, in NDM already after 18 h H1299 cells got roundish and 293 started to detach from the culture plates (Figures 2A and B)." Having observe this phenotype I do not think cell viability analysis of drugs in NDM media and comparing the results of cell viability analysis in BM media is a good approach. For this reason, results section 2.4 is questionable. 

Answer: Thank you for this comment. We re-cultured the detached cells and they attached again to the culture vessels and started proliferation at 18 hours in growth medium (data not shown). This leads us to the suggestion that the cells are still alive and therefore we think that further analysis of cell viability in presence of cytotoxic drugs makes sense. To exclude biased results, we combined all cells (attached and detached cells) for analysis by FACS or metabolic assay. We added this to the manuscript (page 7, lines 306 and following). As shown in figure 5, we didn’t find a significant difference in the cell viability or metabolic activity of cells cultured in BM or NDM. This supports our assumption that detached cells are not dead. 

  1. RNAseq analysis need more clarification. In Figure 3C, why the results were combined for both cell lines? 

Answer: In a first analysis, we assessed differential expression caused by the choice of medium in each cell line independently (Figure 3B). To assess whether some differences could be shown to be cell-line dependent, an interaction term was added to the model. This ensured that the reported differences in the medium effect across the two cell lines (i.e. "cell-line dependent medium effects") were rigorously statistically significant (Figure 3C). This volcano plot should show the differences in the medium effect between H1299 and H1563 cell lines.

We updated the figure legend 3C. We think, it is clear now that this differential expression result reflects how the medium effect differs across H1299 and H1563 cell lines at a statistical level.

  1. Lines 407-412: The results presented here is diverting the authors from main messages of the manuscript.

Answer: We reworked this part completely. We added reasons why we selected the presented genes and conclusions (see page 13). 

Reviewer 3 Report (New Reviewer)

The submitted manuscript entitled by "Interplay between microenvironment and lung adenocarcinoma cell sensitivity to therapies: a spotlight on SREBF1 gene" is well written and deserve publication to this Journal. Authors have done enough work and corrected their manuscript very well. 

 I just suggest to the authors that they have to elaborate conclusion more with future perspectives at proof stage. After that this will be publishable to this Journal. 

Author Response

 just suggest to the authors that they have to elaborate conclusion more with future perspectives at proof stage. After that this will be publishable to this Journal.
Answer: Thank you for your kind comments. We profoundly reworked the conclusion accordingly.

Round 2

Reviewer 2 Report (Previous Reviewer 2)

Authors addressed my concerns.

This manuscript is a resubmission of an earlier submission. The following is a list of the peer review reports and author responses from that submission.

Round 1

Reviewer 1 Report

The study entitled „Interplay between microenvironment and lung adenocarcinoma cell sensitivity to therapies: a spotlight on SREBF1 gene” is very interesting, it uses wide range of methods and collection of patient tissue samples. However, it suffers from some major and minor flaws:

- In the description of Figure 1B authors should add method of staining.

- Figure 1E should be better described, since it is not clear whether DEGs and clusters differ between the lines or between the treatments. What is more, I suggest better description of these very important results presented in Figure 1E, Supplementary Table S3 and Table 1, since we are left only with Table 1, that is hard to follow. Maybe, the expression of cholesterol synthesis related genes should also be presented in form of bar chart?

- What is the rationale to choose POLR2A, ESD and RPS18  as a reference genes in RQPCR? Why different genes were used in different experiments?

- What is the reason, that H1563 cells expressed ~50% viability in BM, as shown in Table 2.?; and after the addition of etoposide their viability increased?

- Figure 2 – what was the rationale to choose “specific genes, which are related to lipid metabolism and LD formation (SREBF1), apoptosis (SERPINE1) and DNA repair processes (p21).”? The number of selected genes seems to be very limited, and I suggest to use at least 3 for each particular pathway.

- What does it mean? “…SREBF1 expression was more widely scattered in tumor samples…”

- Please rephrase this sentence “…On the other hand, neither in NDM nor in BM H1563 cell viability was unaffected by AAT.”

- Please rephrase the Conclusions paragraph, as it is hard to understand: …”The changes in the same cell transcriptome and TME…”

- In general, since the research focuses on SREBF1 expression, it is absolutely needed to show its expression on protein level.

- It is important to answer the question, and in some way to support results shown in Figure 3, – what is the impact of SREBF1 expression on viability of cancer cells?

Reviewer 2 Report

This work describes the impact of microenvironment (LDs, neutrophils and AAT) on cancer cell resistance to chemotherapies. The manuscript is extremely scattered and fail to provide enough evidences in support of the final conclusion. The data presented in the manuscript is very difficult to follow and need a scientific flow. 

Some  comments:

1. "H1299 abundantly 253 formed lipid droplets (LD) even under serum-free conditions, whereas under the same 254 experimental conditions H1563 did not show any traces of LD". The major part of the manuscript is behind this finding. If the hypothesis is true then these cells should varied sensitivity to drugs used. Authors may want to include more in vitro models to conclude anything. 

2. How does artificial editing of microenvironment alters the sensitivity? The manuscript need more data to explain this part. Also, data already presented in needs to be organized. 

3. Subtitles of the results sections should dictate the major findings of each of that section. It is difficult to find what is the final message from each section and how different sections are related to each others. Need flow in the manuscript. 

4. Authors should work to represent data in bar graphs. It is very difficult to follow data in tables. Not readers friendly presentation of results. 

5. Concentrations of drug used is not provided.

6. RNAseq data is not presented properly. Who is compared to what and why- difficult to find. For example- line 276-280 -which cell line?